# Regulation Mechanism for Friction Coefficient of Poly(vinylphosphoric acid) (PVPA) Superlubricity System Based on Ionic Properties

**DOI:** 10.3390/nano12132308

**Published:** 2022-07-05

**Authors:** Mengmeng Liu, Lihui Wang, Caixia Zhang, Yanhong Cheng, Congbin Yang, Zhifeng Liu

**Affiliations:** 1Institute of Advanced Manufacturing and Intelligent Technology, Department of Materials and Manufacturing, Beijing University of Technology, Beijing 100124, China; gdliumengmeng@126.com (M.L.); wanglihui@emails.bjut.edu.cn (L.W.); chengyh@bjut.edu.cn (Y.C.); yangcongbin@bjut.edu.cn (C.Y.); 2Key Laboratory of CNC Equipment Reliability, Ministry of Education, School of Mechanical and Aerospace Engineering, Jilin University, Changchun 130012, China

**Keywords:** poly(vinylphosphonic acid), water-based superlubricity regulation, charge, structure, adsorption, molecular dynamics simulation

## Abstract

Adjustable lubrication aims to achieve active control of the relative motion of the friction interface, providing a new idea for intelligent operation. A new phenomenon of sudden changes of friction coefficient (COF) in the poly(vinylphosphoric acid) (PVPA) superlubricity system by mixing different lubricants, was found in this study. It was found that anions were the critical factor for the COF change. The change degrees of the COF were investigated by a universal micro tribometer (UMT). A quartz crystal microbalance (QCM)-D was used to analyze the adsorption quantity of anions on the PVPA surface. The hydratability of the PVPA interface was controlled by changing the anionic properties (the amount of charge and structure), thus regulating the COF. The adsorption difference of anions is an important reasoning of how anionic properties can regulate the hydratability. It was analyzed by molecular dynamics simulation. For anions carrying different numbers of charges or double bonds, the adsorption quantity of anions was mainly affected by the adsorption degree on the PVPA surface, while the adsorption quantity of anions with different molecular configuration was synergistically regulated by the adsorption degree and adsorption area of anions on the PVPA surface. This work can be used to develop smart surfaces for applications.

## 1. Introduction

Water-based superlubricity not only shows ultra-low friction, but also has the unique advantage of environmental compatibility. Extensive and in-depth studies by scholars have provided much progress in both the macro and micro aspects of water-based superlubricity [1,2,3,4,5,6,7,8,9,10]. With the rapid development of intelligent technology, the demand for responsive surfaces increases, and the control of friction becomes extremely attractive and challenging [11,12,13]. The control of water-based superlubricity has also attracted extensive attention. Polymer-modified surfaces can show exciting superlubricity phenomena [14,15,16,17,18] and environmental response characteristics under certain conditions [19,20,21,22,23], which have great potential to achieve the control of friction.

Salt ions are abundant in the water environment, such as in the ocean and the human body. The study of the salt response of polymer in aqueous solution is of great significance to achieve self-regulation of water-based superlubricity. Understanding and mastering salt response characteristics of the polymer-modified surface is an important prerequisite for achieving friction regulation in a water-based environment. The change of polymer molecular chain properties caused by salt ions is one of the effective ways of controlling the friction behavior of polymer surfaces. Yu et al. proposed that the addition of salts decreases the swelling of poly(2-methacryloyloxyethyl phosphorylcholine) (PMPC) brushes, while the stiffness increases. The increase in stiffening affects the tribo–mechanical response of PMPC brushes, resulting in very low COFs (<0.0004) and a decrease in COF with the anionic size [24]. Raftari et al.’s research pointed out that the addition of larger ions results in a reduced brush thickness in aqueous solution at high salt concentrations, compared with that of no added salt. The relative effect of the different anions follows the Hofmeister series, with $I^{-}$ collapsing the brushes more than ${\mathrm{Br}}^{-}$ and ${\mathrm{Cl}}^{-}$ for the same salt concentration; the tribological properties of the surface changes with the collapse of brushes [25]. The salt response of polymer can cause the change of the interactions between molecular chains and the surface properties as well, thus changing the surface friction properties. Zhang et al. suggested that the addition of ions could change the balance of intra- and intermolecular forces of PMPC molecules, which effectively causes a change in the COFs [26]. Yang et al. synthesized and characterized zwitterionic poly(3-(1-(4-vinylbenzyl)-1H-imidazol-3-ium-3-yl)propane-1-sulfonate) (polyVBIPS) brushes as ion-responsive smart surfaces, which were able to switch to super-low friction surfaces (μ~10^−3^). These salt-responsive surface-switching behaviors between surface friction and lubrication were attributed to cooperative effects of enhanced surface hydration and electrostatic screening [27].

Changing the salt response of a polymer by designing its molecular chain structure is also an effective way to control the COF of the system. Xiao et al. observed a strong dependence of the salt-responsive surface friction/lubrication properties on the intrinsic zwitterionic groups (cationic groups and carbon spacer lengths). It was proposed that such salt-responsive friction behavior became even more pronounced for polymers with an ammonium moiety and longer carbon spacer lengths [28]. Yang et al. developed zwitterionic polymer brushes, which combined a hydrophobic vinylbenzyl group with a zwitterionic imidazolium sulfonate group, increasing the salt-responsive sensitivity. At appropriate ionic conditions, such brushes were able to switch to super-low friction surfaces (μ~10^−3^) [27]. Xiao et al. designed and synthesized a bilayer hydrogel with a pseudo-double-network structure, exhibiting bidirectional bending in response to salt solutions, salt concentrations, and counterion types [23]. It is worth noting that the structure in which polymer chain tethering on the substrate occurred on one end, was easily destroyed by detachment from the chain. A cross-linked structure was introduced by Wu et al. in the form of poly(3-(dimethyl (4-vinylbenzyl) ammonium) propyl sulfonate) (polyDVBAPS) brushes, to offer long-term stability. The responsive behavior of switching between friction and lubrication was obtained by cross-linked polyDVBAPS [29]. It is widely accepted that phosphate groups have a strong affinity to many metal surfaces, and poly(vinylphosphonic acid) (PVPA) is a kind of macromolecule, having a high density of phosphate groups on a polymer backbone [30,31]. Our previous studies showed that PVPA-modified Ti6Al4V has a stable cross-linked network structure, exhibiting superlubricity behavior and a salt-response characteristic [17,32]. Monovalent cations in lubricants reinforce the robustness of the networks by interacting with the PVPA molecules through phosphate anhydrides to prevent their hydrolysis. Anions can regulate the COF of the PVPA interfaces. The water-based superlubricity characteristics of polymer-modified surfaces can be significantly affected by salt anions. However, few studies have involved the regulation of ion properties on friction, and the corresponding mechanism of friction regulation is still unclear.

Considering the theoretical basis of our previous study [17,33], the real-time control of the COF of the PVPA superlubricity system was achieved in this study by introducing different sodium salt solutions in the friction process. The influence of charge, structure and double bond number of anions in sodium solutions on the COF was analyzed, and the relationship between the adsorption number of anions on the PVPA surface and the change degree of the COF was established. The reason for the difference of adsorption of anions with different characteristics on the PVPA surface was further explored via molecular dynamics simulation. To date, very few studies have been performed to investigate the real-time control of the COF via anionic specificity; thus, this work hopefully helps to further the intelligent control of water-based superlubricity.

## 2. Materials and Methods

### 2.1. Materials

PVPA (97%) was provided by Sigma Aldrich (St. Louis, MO, USA). Different kinds of salt (NaCl, NaH_2_PO_4_, Na_2_HPO_4_, Na_3_PO_4_, NaHCO_3_ and Na_2_SO_4_) were purchased from J&K Chemicals (Beijing, China). Ti6Al4V (100 mm × 100 mm) foils with a thickness of 1 mm were supplied by Goodfellow, Inc (Cambridge, UK) and were cut into squares of 10 mm × 10 mm. The method of chemical–mechanical polishing was used in this study. Ti6Al4V was polished by a combination of polishing liquid and polishing pad. The polishing pad was porous polyurethane, and silica sol was used as the base liquid of the polishing liquid. A self-designed polishing fixture was used. In order to make the polishing process stable, a peristaltic pump was used to supply the polishing liquid, to ensure a steady flow. By selecting a reasonable polishing pad, controlling the liquid supply, and adjusting the polishing speed, a smooth and flat polishing surface was obtained. After the polished surface was cleaned, after some time, a bright mirror surface was obtained. Based on the successful application of this polishing method in previous studies [15,16,17,32,33], it was considered that the roughness of all polished samples was around 2 nm. PTFE balls (D ≈ 6 mm) with a roughness of approximately 280 nm, and quartz glass square sheets (10 mm × 10 mm × 1 mm) were purchased from Taobao, Inc (Hangzhou, China). All reagents mentioned above were used as received.

### 2.2. Preparation of PVPA Coatings on Ti6Al4V

PVPA coatings were prepared on Ti6Al4V substrates based on the method of horizontally evaporating self-assembly [17]. First, Ti6Al4V foils with an oxide layer were obtained by heating in air at 140 °C for 8 h. The pretreated foils were then placed horizontally into a PTFE mold. An appropriate PVPA aqueous solution was injected into the mold, and the mold was heated at a temperature of 35 °C to accelerate the physical adsorption of PVPA molecules on the Ti6Al4V. Finally, cross-linked networks were formed on the Ti6Al4V after heating the samples at 260 °C for 6 h.

### 2.3. Evaluation of Tribological Properties

A universal micro-tribometer (UMT-3, Bruker Corporation, Capbell, CA, USA) was used to characterize the tribological properties of the PVPA-modified Ti6Al4V/PTFE interfaces. Briefly, a motor underneath the disk controlled the motor pattern of reciprocation and sliding speed. A precise two-dimensional sensor simultaneously measured the normal load and frictional force generated during sliding contact. PVPA-modified Ti6Al4V and PTFE balls were sampled as tribo-pairs. An amount of 6 mL NaCl solution with a concentration of 0.5 M was used as the base lubricant for the friction experiment, ensuring the PVPA-modified surface was completely immersed. In the base lubrication system, the friction experiments were performed in the reciprocating mode with an initial load of 2.5 N at 25 °C and a reciprocating frequency of 2 Hz, corresponding to a sliding speed of 12 mm/s. The initial contact pressure was approximately 22 MPa. A stable PVPA superlubricity system with a COF between 0.006 and 0.007 was obtained within 5 min. Another sodium salt solution with a concentration of 0.5 M was directly introduced into the base lubricant during the friction process. The volume of salt solution introduced at each time was 2 mL. All experimental results were obtained by averaging the values of at least three repetitions.

### 2.4. Quartz Crystal Microbalance to Explore the Salt Ions Adsorption in Real Time

A quartz crystal microbalance (QCM)-D instrument (Q-sense E4 system, Biolin Scientific, Västra Frölunda, Sweden) was used to measure the adsorption properties of the salt ions. PVPA coatings were prepared on TiO_2_-coated quartz crystal sensors based on the method of horizontally evaporating self-assembly [17]. Then, the TiO_2_-coated quartz crystal sensors with PVPA coating were washed with deionized water. Only the PVPA monolayer was retained on the TiO_2_-coated quartz crystal sensors. 

To simulate the process of introducing different anionic salt solutions into the NaCl solution, NaCl solution with a concentration of 0.5 M was used as the base liquid, and the baseline frequency signal was measured until stable. The salt solution was injected into the sample cell at a flow rate of 0.1 mL/min at 25 °C. Deionized water was subsequently added to rinse the weakly adsorbed ions, and another equilibrium of the frequency was obtained. A new anionic salt solution was further introduced, based on the stable adsorption of Na^+^ and Cl^−^ on the PVPA surface. The changes in both resonance frequency (Δf) and dissipation (ΔD) were simultaneously measured to calculate the adsorption of salt ions on TiO_2_-coated quartz crystal sensors with the PVPA monolayer. According to the change in dissipation (∆D), the dissipation of salt ions was sufficiently small (∆D < ∆f). The salt ions adsorbed on the PVPA monolayer formed a rigid membrane. Compared with the crystal membrane, the rigid membrane was lightweight, thin, and evenly distributed over the active area of the crystal membrane. When the adsorbed film was rigid with a low viscoelasticity, the Sauerbrey equation [34] was used to covert the frequency changes into adsorbed mass changes (Δm), according to the following expression:(1)Δm=−C·Δf/n
where C is a constant of 17.7 ngHz^−1^cm^−2^ which is related to the properties of the quartz crystal, and n is the overtone of the oscillations. In this study, the fifth overtone (n = 5) was selected for computational analysis. 

### 2.5. Molecular Dynamics Simulation

#### 2.5.1. Density Functional Theory (DFT) Study

Density functional theory (DFT) calculations were performed to calculate the interaction energy and to gain insight into the effect of mixing different salts on the PTFE–PVPA system. The simulation was performed using the DFT program DMol3 in Materials Studio (2019) [35]. The physical wave functions were expanded in terms of numerical basis sets, i.e., the Dmol3/GGA-PBE/DNP (3.5) basis set [36]. The core electrons were treated with DFT semi-core pseudopotentials [37]. The exchange-correlation energy was calculated using the Perdew–Burke–Ernzerhof (PBE) generalized gradient approximation (GGA) [38]. A Fermi smearing of 0.005 Ha (1 Ha = 27.211 eV) and a global orbital cutoff of 5.2 Å were employed. The convergence criteria for the geometric optimization and energy calculation were set as follows: (a) a self-consistent field tolerance of 1.0 × 10^−6^ Ha/atom; (b) an energy tolerance of 1.0 × 10^−5^ Ha/atom; (c) a maximum force tolerance of 0.002 Ha/Å; and (d) a maximum displacement tolerance of 0.005 Å.

#### 2.5.2. Model Building

The modeling process of the molecular dynamics simulation was divided into two steps. The origin system was built first. The models of PTFE and PVPA were composed of 1 and 2 monomers, respectively. One Na^+^, one Cl^−^, and two water molecules were placed on the polymer model to form the original system. DFT calculations with the Dmol3/GGA-PBE/DNP basis set were performed to obtain the optimization convergence of energy charge, displacement, and force. The interaction of Cl^−^ with the PVPA chain was then analyzed. The adsorption of salt ions on the PVPA molecular chain were approximately considered as their adsorption on the PVPA surface. After the original system attained stability, one Cl^−^ and one anion were added to it, to build a new system. 

#### 2.5.3. Interaction Energy Calculation

The interaction energy (E_int_), indicating the intensity of the interaction between the components in the system, was derived according to the following equation:(2)$$E_{\mathrm{int}}=E_{\mathrm{tot}}-\sum E_{\mathrm{com}}$$
where $E_{\mathrm{tot}}$ and $E_{\mathrm{com}}$ represent the total energy of the system and the energy of each of its components, respectively. A negative $E_{\mathrm{int}}$ value corresponds to stable adsorption of the components. A more negative value of $E_{\mathrm{int}}$ indicates a stronger interaction in the system.

## 3. Results

### 3.1. Regulation of Charge Quantity on PVPA Superlubricity System

Considering the PVPA superlubricity system obtained in our previous study [16,17], a new method to control the COF in real time during the friction process was found in this study. The COF of the PVPA superlubricity system can be directly regulated by the amount of charge carried by anions in the introducing sodium solutions. In the 0.5 M NaH_2_PO_4_, Na_2_HPO_4_, and Na_3_PO_4_ solutions, $H_{2}{\mathrm{PO}}_{4}^{-}$, ${\mathrm{HPO}}_{4}^{2-}$ and ${\mathrm{PO}}_{4}^{3-}$ have the same structure and the same number of double bonds, while the number of charges increase gradually. The corresponding structures are shown in Figure 1a. The regulations of the COF via anions with different charges are shown in Figure 1b, and the change degrees of the COF are shown in Figure 1c.

The COF of PVPA superlubricity systems all decreased suddenly when anions with different charge were introduced during the friction process. The sudden decrease degree in the COF induced by the introduced $H_{2}{\mathrm{PO}}_{4}^{-}$, ${\mathrm{HPO}}_{4}^{2-}$, and ${\mathrm{PO}}_{4}^{3-}$ were 9.42 ± 0.86%, 25.62 ± 1.51%, and 30.05 ± 2.62%, respectively. The greater the number of charges of phosphate anions, the greater the sudden decrease of degree of the COF for the PVPA superlubricity system. 

The change of interface caused by anions with different charges was an important reason for the sudden decrease in the COF of the superlubricity systems. The more charge the phosphate anion carried, the higher the amount of adsorption that occurred on the PVPA surface. In other words, the larger the increase in the number of charges, the more the COF decreased.

QCM-D measurements were used to determine the number of introduced anions adsorbed on the PVPA surface, and the relationship with the variation of the COF of the PVPA superlubricity system. The real-time adsorption processes of NaH_2_PO_4_, Na_2_HPO_4_, and Na_3_PO_4_, after the stable adsorption of NaCl, are shown in Figure 2a–c, respectively. The adsorbed mass was determined using Equation (1). The introduced cation was identical to the original cation; therefore, the decrease in the frequency was primarily caused by the adsorption of anions with different charges. Based on the obtained adsorption mass and the relative mass of anions with different charges, the approximate amounts of anions adsorbed on the PVPA monolayer were obtained as summarized in Table 1. The adsorption quantities of $H_{2}{\mathrm{PO}}_{4}^{-}$, ${\mathrm{HPO}}_{4}^{2-}$, and ${\mathrm{PO}}_{4}^{3-}$ on PVPA surfaces were 9.51 × 10^15^, 1.55 × 10^16^, and 2.04 × 10^16^ per square centimeter, respectively.

It is proposed that the variation degree of the COF can effectively be regulated by changing the amount of charge carried by the introduced anions. The sudden decrease in the COF was caused by hydrated anion repulsion, which increased the minimum distance between the friction pairs [39]. The repulsion force, which, in turn, influences lubricity, is related to the number of anions adsorbed on the PVPA surface and the capacity of an anion to hold water. The result of the QCM showed that the more charge the introduced phosphate anion carried, the more anions were adsorbed on the PVPA surface. According to Hofmeister series, the more charge an anion carries, the stronger the capacity of the anion to hold water. The more charge an anion carries, the greater the adsorption amount on the PVPA surface and the capacity of an anion to hold water. On this basis, greater repulsive force between friction pairs could be obtained, which resulted in the larger decrease degree of the COF.

### 3.2. Regulation of Ionic Structure on PVPA Superlubricity System

Related studies show that the structure of ions in ionic liquids plays an important role in the regulation of friction [40,41]. However, the effect of the structure of salt ions on the COF has rarely been studied. In our system, it was found that the structure of anions in introduced salt solutions, such as molecular configuration and the number of double bonds, also affected the degree of sudden changes in the COF.

${\mathrm{HCO}}_{3}^{-}$ and $H_{2}{\mathrm{PO}}_{4}^{-}$ carry the same number of charges and double bonds, while the molecular configuration is different due to the difference of central atoms. ${\mathrm{HCO}}_{3}^{-}$ has a configuration of planar, as shown in Figure 3a, while the configuration of $H_{2}{\mathrm{PO}}_{4}^{-}$ is regular tetrahedral, as shown in Figure 1a. The regulation of the COF by anions with different configurations is shown in Figure 3b. For ${\mathrm{HCO}}_{3}^{-}$, the degree of the sudden reduction for the COF was 1.25 ± 1.12%, which was less than for $H_{2}{\mathrm{PO}}_{4}^{-}$ (9.42 ± 0.86%).

When an anion with plane structure was introduced into the base lubricant during the friction process, the COF of the PVPA system was also suddenly reduced. However, compared with the anion with the regular tetrahedral structure, the regulation degree on the COF of the PVPA superlubricity system of the anion with a planar structure was smaller.

${\mathrm{SO}}_{4}^{2-}$ and ${\mathrm{HPO}}_{4}^{2-}$ have the same configuration and the same number of charges. However, the difference of central atoms leads to different numbers of double bonds in ${\mathrm{SO}}_{4}^{2-}$ and ${\mathrm{HPO}}_{4}^{2-}$. There is only one double bond in ${\mathrm{HPO}}_{4}^{2-}$, as shown in Figure 1a, while ${\mathrm{SO}}_{4}^{2-}$ has two double bonds, as shown in Figure 4a. Combined with the regulating degree of ${\mathrm{SO}}_{4}^{2-}$ and ${\mathrm{HPO}}_{4}^{2-}$ on the COF, the regulating effect of the number of double bonds contained in anions on the COF of the PVPA superlubricity system was analyzed. The result is shown in Figure 4b. When ${\mathrm{SO}}_{4}^{2-}$ or ${\mathrm{HPO}}_{4}^{2-}$ was introduced during the friction process, the COFs suddenly decreased by 11.82 ± 2.02%, and 25.62 ± 1.51%, respectively. It can be assumed that the lower the number of double bonds in the introduced anion, the greater the degree of the sudden decrease in the COF.

The real-time adsorption processes of ${\mathrm{HCO}}_{3}^{-}$ and ${\mathrm{SO}}_{4}^{2-}$ on the PVPA surface are shown in Figure 5a and Figure 5b, respectively. The adsorption quantity of ${\mathrm{HCO}}_{3}^{-}$ was 8.18 × 10^15^ per square centimeter, smaller than that of $H_{2}{\mathrm{PO}}_{4}^{-}$ (9.51 × 10^15^ per square centimeter). Considering such results, one conclusion can be drawn as follows: the regular tetrahedral structure is more conducive to the adsorption of anions on the PVPA surface than the plane structure. In addition, the adsorption quantity of ${\mathrm{SO}}_{4}^{2-}$ was 8.21 × 10^15^ per square centimeter, which was smaller than that of ${\mathrm{HPO}}_{4}^{2-}$ (1.55 × 10^16^ per square centimeter). It can be considered that the more double bonds the anions contain, the fewer number of anions are adsorbed on the surface. 

Anions with different structures that were introduced to the friction process could all suddenly reduce the COF of PVPA superlubricity system. This was due to the introduction of new anions to enhance the hydratability of the PVPA network interfaces. The degree of variation of COF was closely related to the structure of the introduced anions. The degree of change in the hydratability of the PVPA interface was determined by the capacity of an introduced anion to hold water and the amount of introduced anions adsorbed on the PVPA interface. As ${\mathrm{HCO}}_{3}^{-}$ and $H_{2}{\mathrm{PO}}_{4}^{-}$ have similar capacity to hold water, the main reason for the change difference of COF was the different adsorption quantity of introduced anions on the PVPA interface. In other words, the molecular configuration of the anions regulated the number of anions adsorbed on the PVPA surface. It is worth noting that the capacity of ${\mathrm{SO}}_{4}^{2-}$ to hold water is larger than that of ${\mathrm{HPO}}_{4}^{2-}$, but the sudden reduction in COF caused by ${\mathrm{SO}}_{4}^{2-}$ was less than that caused by ${\mathrm{HPO}}_{4}^{2-}$. It can be concluded that for anions with different numbers of double bonds, the adsorption quantity on the PVPA surface plays a greater role than the capacity to hold water, for the sudden reduction in COF. The result of the QCM showed that the more double bonds an anion contained, the less quantity of anions were adsorbed on the PVPA surface. Changing the number of anionic double bonds to control the number of anions adsorbed on the surface, and then changing the degree of sudden change of COF, was also an effective way to achieve accurate control of COF.

### 3.3. Simulation Mechanism for the Difference of Anionic Adsorption

The results in Section 3.1 and Section 3.2 show that the adsorption number of introduced anions on the PVPA surface was an important factor affecting the sudden decrease in COF. Adjusting the amount of charge or the structure of an anion could change the number of anions adsorbed on PVPA surface, and then regulate the sudden decreased degree of COF. Molecular dynamics simulation was used in this section to calculate the interaction between different anions and the PVPA surface. The reason for the difference of the number of different anions adsorbed on the PVPA surface was further analyzed.

The interactions between the PVPA molecular chain, and anions before and after the introduction of different anions, are shown in Figure 6. The influence of cations can be ignored when investigating the effects of the introduced anions. To more clearly reflect the interactions between the introduced anions and PVPA chains, and those between Cl^−^ and the PVPA chains before and after the introduction, Na^+^ and water molecules were removed from the new system. Notably, the removal behavior did not affect the specific value of the interaction energy.

Before and after the introduction of different anions, the interaction energies between Cl^−^ and the PVPA molecular chain were recorded in Table 2. Through the analysis of data, it could be seen that the introduction of another anion, on the basis of the stability of the original system, caused a certain disturbance to the adsorption state of Cl^−^ on the PVPA surface, but this disturbance had no direct influence on the degree of the sudden decrease in the COF.

The interaction energies between the introduced anions and the PVPA molecular chain are shown in Figure 7. When the introduced anions were $H_{2}{\mathrm{PO}}_{4}^{-}$, ${\mathrm{HPO}}_{4}^{2-}$ and ${\mathrm{PO}}_{4}^{3-}$, the interaction energies with PVPA chain were 0.20, −98.46, and −125.89 Kcal/mol, respectively. When the number of anion charges changed from 1 to 2, the interaction between the introduced anion and PVPA surface changed from repulsion to adsorption. In addition, the adsorption degree of the introduced anion increased with the increase in charge. Notably, the change of the interaction energy between the introduced anion and PVPA surface when the number of the anion charge changed from 1 to 2, was much larger than the change from 2 to 3. The minimum distances of $H_{2}{\mathrm{PO}}_{4}^{-}$, ${\mathrm{HPO}}_{4}^{2-}$, and ${\mathrm{PO}}_{4}^{3-}$ to the PVPA molecular chain were 3.366, 1.025, and 0.998Ȧ, respectively, as shown in Figure 6b–d. It can be speculated that the oxygen atoms containing a lone pair electrons in ${\mathrm{HPO}}_{4}^{2-}$ and ${\mathrm{PO}}_{4}^{3-}$ form hydrogen bonds with the hydrogen atoms in the phosphate radical at the end of the PVPA molecular chain, which greatly enhances the interaction energy of ${\mathrm{HPO}}_{4}^{2-}$ and ${\mathrm{PO}}_{4}^{3-}$ with the PVPA molecular chain. The formation of the hydrogen bond causes the interaction between the anion with one negative charge and the PVPA surface to obviously be different to that of the anion with two or three negative charges. In addition, ${\mathrm{HPO}}_{4}^{2-}$ has two oxygen atoms with a lone pair of electrons that can form hydrogen bonds with hydrogen atoms in the phosphate radical at the end of the PVPA molecular chain, while ${\mathrm{PO}}_{4}^{3-}$ has three oxygen atoms with a lone pair of electrons that can form hydrogen bonds. Therefore, the adsorption degree of ${\mathrm{PO}}_{4}^{3-}$ on the PVPA surface is greater than that of ${\mathrm{HPO}}_{4}^{2-}$, which is indirectly indicated by the shortest distance between the introduced anion and the PVPA chain shown in Figure 6c,d. For introduced anions with different numbers of charges, the formation of the hydrogen bond will obviously increase the adsorption degree of the anion on the surface. The larger the degree of adsorption of anions on the surface, the higher the adsorption number, and the greater the sudden decrease in degree of the COF.

When the introduced anion was ${\mathrm{HCO}}_{3}^{-}$, its interaction energy with the PVPA chain was −14.67 Kcal/mol. The minimum distance of ${\mathrm{HCO}}_{3}^{-}$ to the PVPA chain was 1.532 Ȧ, as shown in Figure 6e. The oxygen atom attached to the double bond is likely to form hydrogen bonds with the hydrogen atom in the phosphoric acid at the end of the PVPA chain. The formation of the hydrogen bond causes the interaction energy of ${\mathrm{HCO}}_{3}^{-}$ with the PVPA molecular chain to be larger than that of $H_{2}{\mathrm{PO}}_{4}^{-}$ with the PVPA molecular chain. In other words, the adsorption degree of ${\mathrm{HCO}}_{3}^{-}$ on the PVPA surface is greater than that of $H_{2}{\mathrm{PO}}_{4}^{-}$. Theoretically, the adsorption amount of ${\mathrm{HCO}}_{3}^{-}$ on the PVPA surface should be larger than that of $H_{2}{\mathrm{PO}}_{4}^{-}$, while the result was just the opposite. Since ${\mathrm{HCO}}_{3}^{-}$ and $H_{2}{\mathrm{PO}}_{4}^{-}$ have the same number of charges and double bonds, such contrary conclusion is speculated to be caused by the structural difference of the anions. ${\mathrm{HCO}}_{3}^{-}$ is a planar structure. The oxygen atoms at the end of the double bond are 2.253 and 2.285Ȧ away from the other two oxygen atoms. The distance between the oxygen atom at the end of the C-O single bond and the hydrogen atom is 2.226 Ȧ. The length of the O-H single bond is 0.975 Ȧ. Assuming that an ${\mathrm{HCO}}_{3}^{-}$ is completely adsorbed on the PVPA surface, its corresponding adsorption area is 0.0330 nm^2^. The structure of $H_{2}{\mathrm{PO}}_{4}^{-}$ is regular tetrahedral, and the adsorption area *S* of a single $H_{2}{\mathrm{PO}}_{4}^{-}$ can be calculated as follows:(3)$$S=\frac{\surd 3}{4}*{\left(2L*sin\frac{\theta }{2}\right)}^{2}$$
where *L* = 0.16 nm is the bond length of P-O bond, and *θ* = 109.5° is the bond angle between O–P–O. The adsorption area of an $H_{2}{\mathrm{PO}}_{4}^{-}$ on the PVPA surface is 0.0296 nm^2^. Under the combined action of the adsorption degree and adsorption area of anions on the PVPA surface, the adsorption amount of ${\mathrm{HCO}}_{3}^{-}$ on PVPA surface is less than that of $H_{2}{\mathrm{PO}}_{4}^{-}$, resulting in the sudden decrease in degree of the COF induced by ${\mathrm{HCO}}_{3}^{-}$ to be less than that of $H_{2}{\mathrm{PO}}_{4}^{-}$.

When the introduced anion was ${\mathrm{SO}}_{4}^{2-}$, its interaction energy with PVPA chain was −1.24 Kcal/mol, as shown in Figure 7. In contrast to ${\mathrm{HPO}}_{4}^{2-}$, ${\mathrm{SO}}_{4}^{2-}$ contains two double bonds due to central atomic differences. It can be speculated that the higher the number of double bonds in anions, the lower the interaction between anions and the PVPA molecular chain, the lower the adsorption number of anions on the PVPA surface, and the smaller the decrease in the degree of the COF.

The results showed that for anions carrying different numbers of charges, the more charge they carry, the more likely they are to form hydrogen bonds with the PVPA surface, which leads to greater interaction between anions and the PVPA surface, and the more they adsorb on the PVPA surface. For anions with different structures, the adsorption number of anions on the PVPA surface is synergistically regulated by the interaction degree between anions and the PVPA surface, and the adsorption area on the PVPA surface. In addition, the adsorption quantity of anions with different numbers of double bonds on the PVPA surface is mainly affected by the degree of adsorption on the PVPA surface.

## 4. Conclusions

Based on the water-based superlubricity system obtained in previous studies, a new phenomenon, that the COF of the PVPA superlubricity system changes suddenly via the addition of other sodium salts (NaH_2_PO_4_, Na_2_HPO_4_, Na_3_PO_4_, NaHCO_3_ and Na_2_SO_4_) to the basic lubricant (0.5 M NaCl solution) during the friction process, was found in this study. The response of the PVPA coating to anions with different characteristics is real-time, and the response degree can be quantified as the change degree of the COF, which can be accurately calculated and controlled. Phenomena and mechanisms of the sudden change of the COF induced by anionic properties (the number of charges and the structure) were investigated. 

The introduction of anions in the friction process enhances the hydratability of the PVPA interface, resulting in a sudden decrease in the COF. The hydratability of the PVPA interface is co-determined by the capacity of the introduced anion to hold water and the adsorption amount of the introduced anions on the PVPA surface, which can be regulated by the properties of introduced anions. Thus, the degree of sudden reduction in friction coefficient can be adjusted by the properties of introduced anions. It is proposed that the increase in the charge carried by the anion is favorable for it to form hydrogen bonds with the hydroxyl group on the PVPA molecular chain, which will enhance the adsorption degree of anions on the PVPA surface, leading to an increase in the number of anions adsorbed on the PVPA surface. In addition, the more charge an anion carries, the better its capacity to hold water. Thus, the degree of the sudden reduction in COF can be enhanced by increasing the number of charges of the introduced anion. For introduced anions with different structures, the influence of adsorption quantity on the change of COF is greater than the capacity of an anion to hold water. It is noteworthy that the number of anions with different molecular configuration adsorbed on the PVPA surface is synergistically regulated by the adsorption degree and the adsorption area of anions on the PVPA surface. Additionally, the more double bonds an anion carries, the smaller the adsorption degree of the anion on the PVPA surface, and the lower the number of the adsorption number of anions. We will conduct a detailed study on the degree of regulation of the lubricant mixing ratio on COF in the following study, perfecting the superlubricity regulation system based on salt ions. These studies will provide a good theoretical basis for adaptive control of water-based superlubricity.

## Figures and Tables

**Figure 1 nanomaterials-12-02308-f001:**
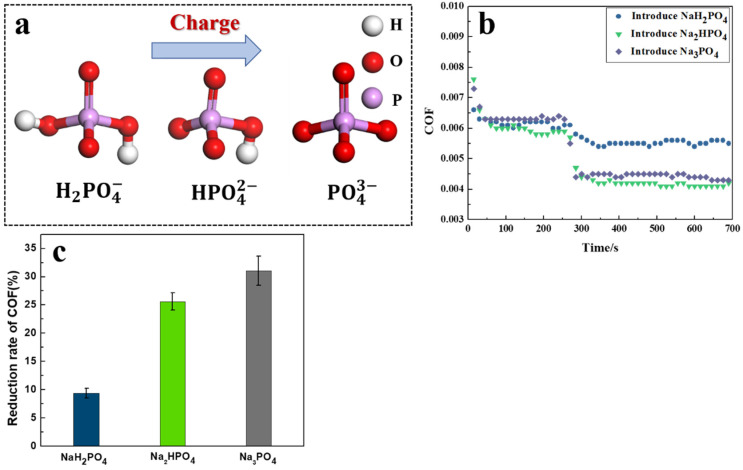
(**a**) Schematic diagram of the structures of anions with different charges, (**b**) Variations in the COF of PVPA-modified Ti6Al4V in a NaCl solution sliding against PTFE balls induced by the introduction of anions with different charges, (**c**) Reduction rate of COF.

**Figure 2 nanomaterials-12-02308-f002:**
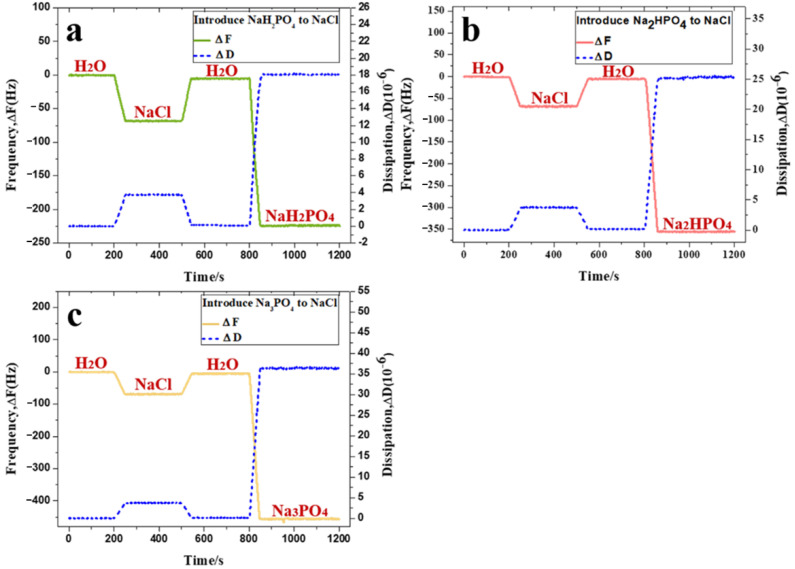
Adsorption behavior of salt ions measured using the QCM: (**a**) NaH_2_PO_4_, (**b**) Na_2_HPO_4_, and (**c**) Na_3_PO_4_.

**Figure 3 nanomaterials-12-02308-f003:**
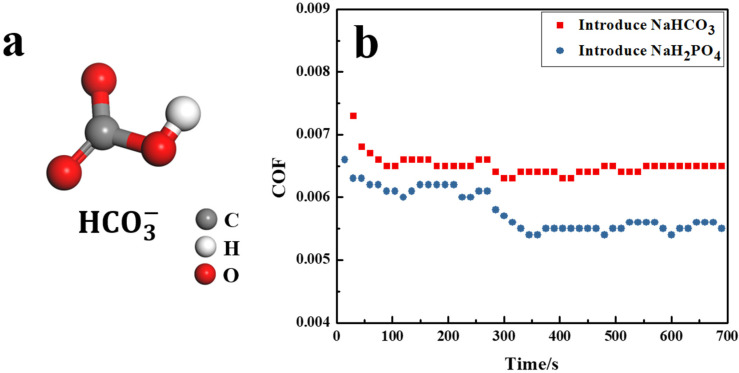
(**a**) Schematic diagram of the structure of ${\mathrm{HCO}}_{3}^{-}$, (**b**) Variations in the COF of PVPA-modified Ti6Al4V in a NaCl solution sliding against PTFE balls induced by the introduction of anions with different structures.

**Figure 4 nanomaterials-12-02308-f004:**
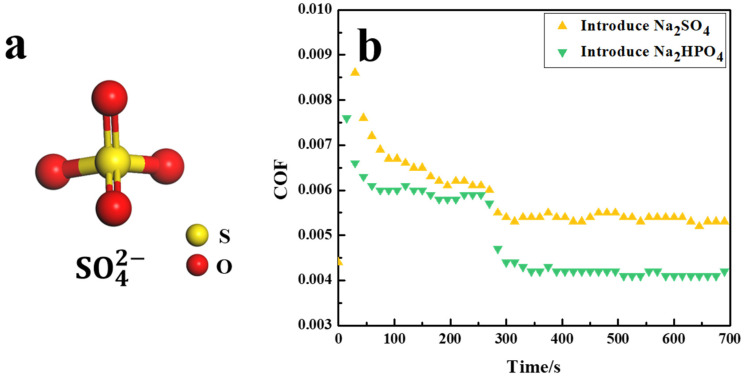
(**a**) Schematic diagram of the structure of ${\mathrm{SO}}_{4}^{2-}$, (**b**) Variations in the COF of PVPA-modified Ti6Al4V in a NaCl solution sliding against PTFE balls induced by the introduction of anions with different number of double bonds.

**Figure 5 nanomaterials-12-02308-f005:**
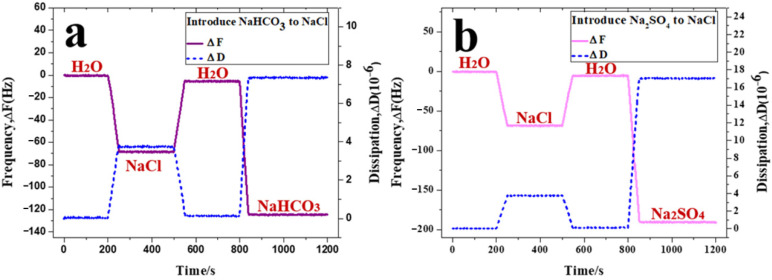
Adsorption behavior of (**a**) NaHCO_3_ and (**b**) NaSO_4_, measured using the QCM.

**Figure 6 nanomaterials-12-02308-f006:**
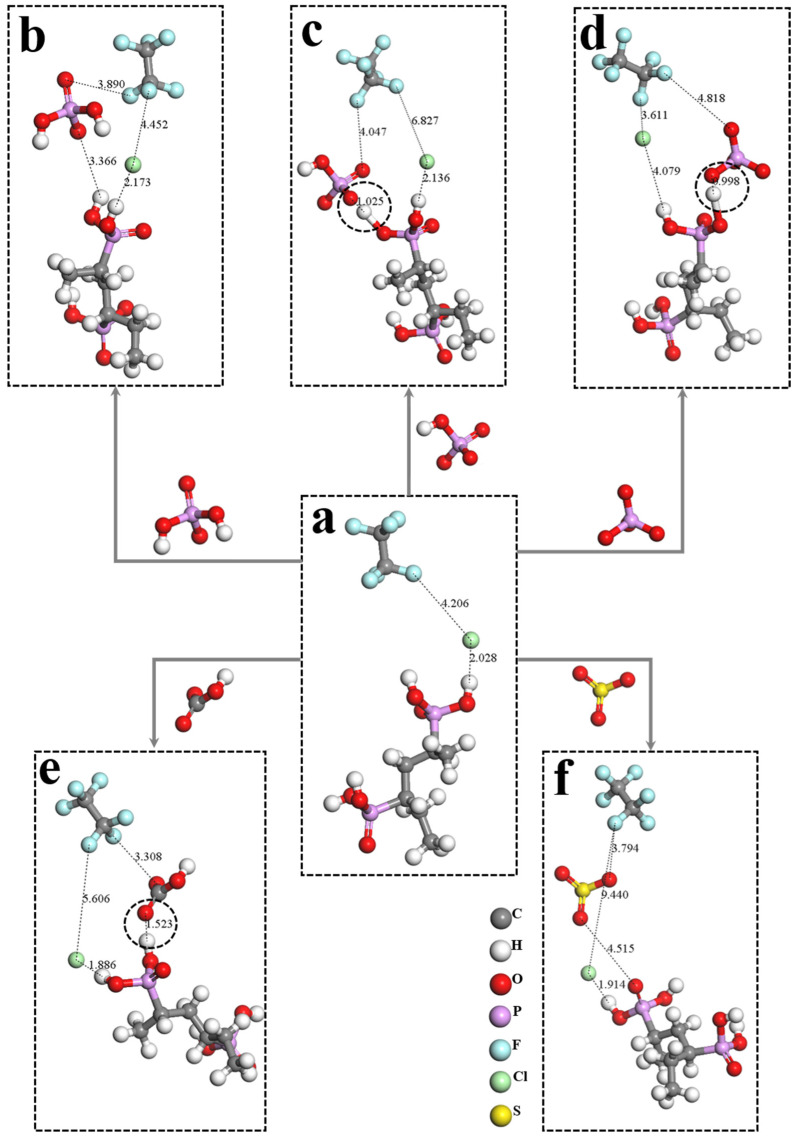
Intermolecular distance between the introduced anions and the PVPA chain and that between Na+ and the PVPA chain before and after the introduction of the anion: (**a**) Cl^−^-PVPA, (**b**) Cl^−^-PVPA and $H_{2}{\mathrm{PO}}_{4}^{-}$-PVPA, (**c**) Cl^−^-PVPA and ${\mathrm{HPO}}_{4}^{2-}$-PVPA. The black dashed circle in (**c**) was the hydrogen bond between hydrogen and oxygen atom, (**d**) Cl^−^-PVPA and ${\mathrm{PO}}_{4}^{3-}$-PVPA. The black dashed circle in (**d**) was the hydrogen bond between hydrogen and oxygen atom, (**e**) Cl^−^-PVPA and ${\mathrm{HCO}}_{3}^{-}$-PVPA. The black dashed circle in (**e**) was the hydrogen bond between hydrogen and oxygen atom (**f**) Cl^−^-PVPA and ${\mathrm{SO}}_{4}^{2-}$-PVPA.

**Figure 7 nanomaterials-12-02308-f007:**
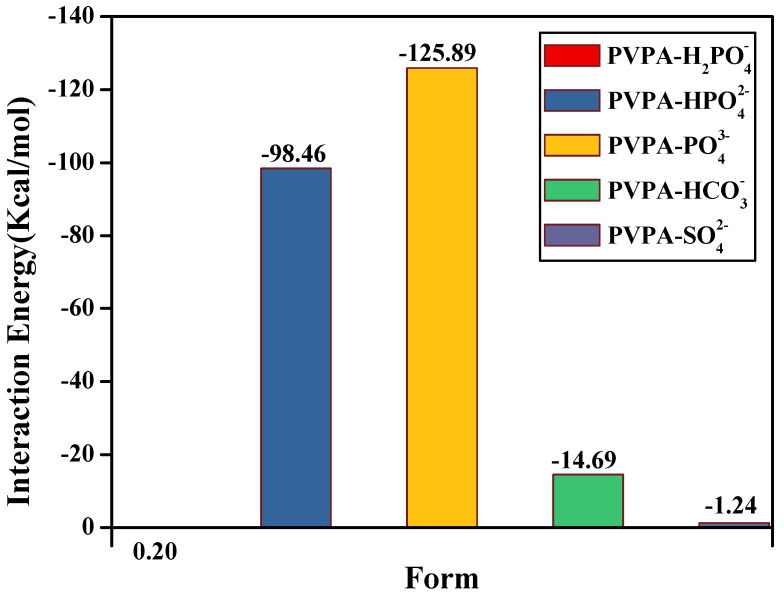
Energies of the interactions of the introduced anions with the PVPA chain.

**Table 1 nanomaterials-12-02308-t001:** Adsorption results of anions with different charges.

Introduced Anions	Adsorption	
	Mass (ng/cm^2^)	Number
$H_{2}{\mathrm{PO}}_{4}^{-}$	773.49	9.51 × 10^15^
${\mathrm{HPO}}_{4}^{2-}$	1239	1.55 × 10^16^
${\mathrm{PO}}_{4}^{3-}$	1593	2.04 × 10^16^

**Table 2 nanomaterials-12-02308-t002:** Calculated interaction energies of the PVPA chain with Cl^-^ before and after the introduction of anions.

Solution	Form	Interaction Energy (Kcal/mol)
NaCl	$\mathrm{PVPA}\text{-}{\mathrm{Cl}}^{-}$	−13.98
Introduce NaH_2_PO_4_ to NaCl	$\mathrm{PVPA}\text{-}{\mathrm{Cl}}^{-}$	−11.75
Introduce Na_2_HPO_4_ to NaCl	$\mathrm{PVPA}\text{-}{\mathrm{Cl}}^{-}$	−12.09
Introduce Na_3_PO_4_ to NaCl	$\mathrm{PVPA}\text{-}{\mathrm{Cl}}^{-}$	−6.22
Introduce NaHCO_3_ to NaCl	$\mathrm{PVPA}\text{-}{\mathrm{Cl}}^{-}$	−15.41
Introduce Na_2_SO_4_ to NaCl	$\mathrm{PVPA}\text{-}{\mathrm{Cl}}^{-}$	−14.78

## Data Availability

Not applicable.

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
