# Peer review of "Regulation Mechanism for Friction Coefficient of Poly(vinylphosphoric acid) (PVPA) Superlubricity System Based on Ionic Properties"

_nanomaterials, 2022, doi:10.3390/nano12132308_

Round 1

Reviewer 1 Report

The paper by Liu and co-authors report the results of measurements of the coefficient of friction of lubricated PVPA-modified Ti6Al4V/PTFE interfaces, demonstrating controllable friction by mixing different lubricants.

The authors used a micro-tribometer to study the real-time change of the coefficient of friction as different kind of salts where added to solution.  They concluded that the coefficient of friction mainly depends on the degree of hydratability of the PVPA surface, which, in turn, is determined by the properties of the different adsorbed anions. Specifically, based on QCM measurements of the degree of adsorption and on the insights provided by DFT-based geometry optimization of model interfaces the authors highlighted the role of (1) different number of anionic charge and (2) anionic molecular configuration affecting the degree of hydratability.

The research was properly designed and conducted, the paper is well written and suitable for publication in nanomaterials.

Author Response

Thank you for your review.

Reviewer 2 Report

The topic of the manuscript: “Regulation mechanism for friction coefficient of poly(vi- 2 nylphosphoric acid )(PVPA) superlubricity system based on 3 ionic properties” is very interesting and very timely.

However, there are small shortcomings that should be refined (see below).

Page 2, line 55: Zhang et al – a period is missing, it should be: Zhang et al.

Page 2, line 57: Yang et al  - a period is missing (same like above), “al” is always followed by a period. In the sentences below the same - please correct in the text

Page 3, line 106: How these Ti6Al4V foils were polished? Mechanically? Chemically? Please include this information in the text. The way of polishing is very important and could affect the properties of the surface (e.g. adhesion of the coatings) and affect the results of modifications. How their roughness was measured (all the samples)? Please give this information in the text.

Page 5, line 204: Figure 1 b) - since the results are the average of at least 3 experiments why there are no standard deviations on the graph?

I recommend this work to publication.

Author Response

Firstly, I would like to express my thanks for your thoughtful comments which are valuable for the improvement of our manuscript. Based on your comments, careful revisions have been made to the revised manuscript. Within the manuscript, the text where key revisions have been made is marked in red. The following text is point-by-point responses to your comments.
